# Genome Sequencing of Consanguineous Family Implicates Ubiquitin-Specific Protease 53 (*USP53*) Variant in Psychosis/Schizophrenia: Wild-Type Expression in Murine Hippocampal CA 1–3 and Granular Dentate with AMPA Synapse Interactions

**DOI:** 10.3390/genes14101921

**Published:** 2023-10-09

**Authors:** Ambreen Kanwal, Sohail A. Sheikh, Faiza Aslam, Samina Yaseen, Zachary Beetham, Nathan Pankratz, Connie R. Clabots, Sadaf Naz, José V. Pardo

**Affiliations:** 1School of Biological Sciences, University of the Punjab, Lahore 54590, Pakistan; ambreen.phd.sbs@pu.edu.pk (A.K.); faizaaslam.phd.sbs@pu.edu.pk (F.A.); samyyaseen@gmail.com (S.Y.); 2Cognitive Neuroimaging Unit, Minneapolis Veterans Health Care System, Minneapolis, MN 55417, USA; 3Department of Psychiatry, University of Minnesota, Minneapolis, MN 55454, USA; 4Department of Psychiatry, Hawkes Bay Hospital, Hastings 4120, New Zealand; sheik004@hotmail.com; 5Division of Computational Pathology, Department of Laboratory Medicine and Pathology, University of Minnesota, Minneapolis, MN 55455, USA; beetham63@gmail.com (Z.B.);; 6Medicine Patient Service Line, Minneapolis Veterans Health Care System, Minneapolis, MN 55417, USA; connie.clabots@va.gov

**Keywords:** deubiquitinating enzyme (DUB), intercellular junction, GRIA2, GRIP2, homozygosity mapping

## Abstract

Psychosis is a severe mental disorder characterized by abnormal thoughts and perceptions (e.g., hallucinations) occurring quintessentially in schizophrenia and in several other neuropsychiatric disorders. Schizophrenia is widely considered as a neurodevelopmental disorder that onsets during teenage/early adulthood. A multiplex consanguineous Pakistani family was afflicted with severe psychosis and apparent autosomal recessive transmission. The first-cousin parents and five children were healthy, whereas two teenage daughters were severely affected. Structured interviews confirmed the diagnosis of DSM-V schizophrenia. Probands and father underwent next-generation sequencing. All available relatives were subjected to confirmatory Sanger sequencing. Homozygosity mapping and directed a priori filtering identified only one rare variant [MAF < 5(10)^−5^] at a residue conserved across vertebrates. The variant was a non-catalytic deubiquitinase, *USP53* (p.Cys228Arg), predicted in silico as damaging. Genome sequencing did not identify any other potentially pathogenic single nucleotide variant or structural variant. Since the literature on *USP53* lacked relevance to mental illness or CNS expression, studies were conducted which revealed USP53 localization in regions of the hippocampus (CA 1–3) and granular dentate. The staining pattern was like that seen with GRIA2/GluA2 and GRIP2 antibodies. All three proteins coimmunoprecipitated. These findings support the glutamate hypothesis of schizophrenia as part of the AMPA-R interactome. If confirmed, *USP53* appears to be one of the few Mendelian variants potentially causal to a common-appearing mental disorder that is a rare genetic form of schizophrenia.

## 1. Introduction

Psychosis is a prevalent (e.g., 1–4% Danish general population [1]) symptom complex consisting of hallucinations, delusions, and thought disorder. Schizophrenia, the quintessential psychotic disorder, is also common (1%) and begins typically in teenage/early adulthood; it is widely considered a neurodevelopmental disorder. Despite the clinical importance of psychosis/schizophrenia, treatments remain limited. Developed serendipitously decades ago, treatments have significant morbidity and mortality. Rational drug design is not possible in the absence of pathophysiology, impeding clinical translation.

Genetics is the most straight forward path from disease to its pathophysiology. The genetic architecture of most medical disorders lies in a continuum from common low-risk variants with small effects to rare high-risk variants with large effects on phenotype [2]. However, evidence for human Mendelian genes (i.e., simple inheritance, large effects on phenotype, and rarity) in psychiatry remains limited [3].

Rare variants may play a role in the risk for some common neuropsychiatric presentations. In schizophrenia, significantly greater ultra-rare missense or loss-of-function (LoF; severe disruption of protein coding gene by decreased expression or impaired functions) variants occur in founder populations such as Ashkenazi Jews [4]. A high burden of rare LoF variants and damaging missense variants in genes intolerant to mutation surfaced in schizophrenia [5]. Risk genes for schizophrenia are enriched for ultra-rare coding variants particularly in genes associated with the development and regulation of the structure and function of synapses—especially glutamatergic NMDA and AMPA receptors [6]. In patients with bipolar disorder either with or without psychosis, whole exome sequencing also revealed several ultra-rare protein-truncating variants affecting mutation-intolerant genes [7]. Several rare protein-truncating variants among evolutionarily conserved genes were associated with schizophrenia and were largely consistent across five ancestral populations [8]. In summary, both common and rare risk genes associated with schizophrenia were reviewed recently [9].

A useful approach for the discovery of rare Mendelian variants is the study of pedigrees in which multiple affected progenies are born to unaffected consanguineous parents suggesting recessive inheritance. Although Mendelian disorders are usually rare given the negative selection pressure, they often can appear common at the phenotypic or syndromal level based on genetic heterogeneity. Different genes can associate with the same phenotype requiring molecular definition for nosology [10].

In this context, identification of a unique disease is based on the family unit because coinheritance of two very rare genetic variants is extremely unlikely. So, a rare variant inherited in a consanguineous pedigree defines the disorder genetically and its familial phenotypic spectrum. Unlike the candidate gene approach leading frequently to false associations based on a limited understanding of the relevant biology [11], an unbiased exploration for variants of interest (VOIs) requires a directed search on the whole genome filtering out variants of little interest based on a priori criteria and on inheritance within the particular pedigree.

Here, we identify in a multiplex consanguineous family affected by psychosis (diagnostically DSM-V schizophrenia), a novel missense variant in a protein previously undetected in the CNS and unassociated with any mental disease. The wild-type protein is expressed in the hippocampus and is part of the interactome of the AMPA synapse. These findings are consistent with dominant etiologic theories, albeit indirect, about schizophrenia.

## 2. Methods

### 2.1. Subjects

Written informed consent was obtained from Pakistani participants after detailed explanation in Urdu and with ethical review at School of Biological Sciences (IRB# 00005281, FWA 00010252). A family, PSY01 (Figure 1A), identified at the outpatient department of a local hospital, had two siblings with severe schizophrenia. Hearing was assessed normal by observation (see below); the subjects refused audiometry. The family denied history of jaundice during proband’s infancy or childhood (see below). Standard rating scales and structured interviews were used: Diagnostic Interview for Genetic Studies, DIGS [12]; Diagnostic Interview for Psychosis and Affective Disorders, DI-PAD [13]; Positive and Negative Syndrome Scale, PANSS [14]; Hamiltonc Depression and Anxiety Rating Scale, HAM-D [15] HAM-A [16]; and modified MiniMental Status Examination; mMINI. All available and willing affected and unaffected family members were interviewed. The complete interviews lasted approximately 3.5 h. Additional details about participant recruitment, assessment, interviews, and other special circumstances are published elsewhere [17] as well as in the Appendix A.

### 2.2. Genetics: Sequencing and Analysis

Whole-exome sequencing, variant filtering, and analyses were performed for the two patients and their father as described previously [18], and the exome data were mapped to GRCh37/hg19 genome assembly. Variants were prioritized based on the following filters: (1) rarity (MAF < 0.01). (2) in silico predicted pathogenicity as damaging; (3) absence as homozygote in psychiatrically normal volunteers; (4) missense occurrence in an evolutionarily conserved amino acid; and (5) not polymorphic in the indigenous population. The candidate variant had to show autosomal recessive segregation within the pedigree. Segregation in family was confirmed by Sanger sequencing of the specific amplified products from DNA of all participants using pertinent primers (Appendix A). Absence of VOI as a common polymorphism from the local population was determined by allele specific PCR (Appendix A and Methods) on DNA of 400 unrelated ethnically matched unaffected individuals (800 chromosomes).

Whole genome sequencing (WGS) was performed on the two affected girls and their father (Macrogen, Seoul, South Korea). It used the Illumina (San Diego, CA, USA) pipeline with fragmentation of 100 ng of genomic DNA using adaptive focused acoustic technology (AFA; Covaris, Woburn, MA, USA). TrueSeq was used for creation of the library. The fragmented DNA was end-repaired to create 5′-phosphorylated, blunt-ended, double-stranded DNA molecules. Following end-repair, DNA was selected for size with a magnetic bead-based method. These DNA fragments went through the addition of a single ‘A’ base, and ligation of indexing adapters. The purified libraries were quantified using qPCR according to the qPCR Quantification Protocol Guide (KAPA Library Quantification kits for Illumina Sequencing platforms) and qualified using the Agilent Technologies 2200 TapeStation (Agilent, Santa Clara, CA, USA). WGS analysis followed the GATK pipeline with variant selection detailed further below.

The general GATK workflow pipeline was used for discovery of germline variants (Appendix A). This analysis used the standard GATK models and statistics with default parameter values [19]. This pipeline maps the WGS raw reads to the GRCh38/hg38 reference genome producing an aligned file (sam/bam) sorted by coordinates. Duplicate reads were marked to mitigate biases from procedures such as PCR amplification. The base quality scores, critical for variant calling algorithms, require recalibration based on covariate measurements of calls in the dataset and in each sample. This recalibration addresses biases from sample preparation or preprocessing of WGS raw reads after mapping to GRCh38/hg38 reference genome.

The variant call files (vcf) cof the three samples were jointly genotyped and filtered for all SNPs and for use subsequently as input to VerifyBamID. This program calculates a “freemix score”, a metric of sample contamination. The freemix scores (0.16744, 0.15962, 0.16012) were above the typical threshold of <0.05 used for larger datasets. The limited data prevented recalibration or tranching for technical artifacts. To evaluate QC further, the vcf files were filtered again leaving only common variants found in public databases. This manipulation normalized the freemix scores indicating acceptable quality and lack of contamination. Additionally, peddy was used to check demographics for genetic relatedness, ancestry, and sex. One daughter (IV:3) was incorrectly flagged as male, but on closer inspection, she just missed the cutoff for female assignment. Also, the cumulative proportion of X- and Y-linked reads clearly supported female assignment.

All variants in the joint-genotyped vcf file were annotated for functional impact (putative amino acid changes) using SnpEff, and only variants that led to a loss of function mutation (high impact) or missense mutation (moderate impact) were considered. In addition, variants in genes known to have a heavy mutational load in the population were excluded based on the GDI score [20]: the gene damage index (GDI) is a metric of the accumulated mutational damage of each human gene in the healthy human population based on the 1000 Genomes Project database. These GDI scores for the genes containing our VOIs are included in Appendix A.

While our primary search was limited to variants inherited in an autosomal recessive fashion, we considered a total of five genetic models. (1) Autosomal Dominant or Multiple Hits: Genes with two high impact (i.e., loss-of-function) variants; allelic depth (AD) > 20; genotype quality (GQ) > 50; and gene damage index (GDI) < 75 percentile. (2) Variants of Interest (VOI) or previously reported genes: one high impact and one moderate impact (e.g., missense); AF < 0.005; no quality filters. (3) Autosomal Recessive: one high impact and one moderate impact variant; AD > 20; GQ > 50; AF < 0.005; and GDI < 75 percentile. (4) X-linked: Autosomal Recessive: one high impact and one moderate impact homozygous X-linked variant; AD > 20; GQ > 50; AF < 0.005; and GDI < 75 percentile. (5) Compound Heterozygote: One or more high impact variants and one or more moderate impact variants in the same sample on separate haplotypes as determined by IGV. Each variant was then manually reviewed using IGV. All reported variants passed our within-laboratory threshold for standard quality criteria and were checked against existing databases (gnomAD, BRAVO). Variants were reported if the number of calls matched the corresponding requirements of their source (i.e., genetic model).

Structural variant discovery (Appendix A) was performed using Model free analysis. Structural variants (SVs), including copy number variants (CNVs), inversions, and translocations, were called using the Lumpy algorithm implemented in smoove using the aligned reads (bam files) after pre-processing as above. CNVs were filtered using duphold finding 23 deletions (duplications) with <0.7-fold change (>1.3-fold change, respectively) for variant depth relative to flanking regions or for genomic regions with similar GC content. AnnotSV was used for annotation of the filtered SVs and for assignment to a category of pathogenicity: benign, likely benign, variant of unknown significance, likely pathogenic, and pathogenic. Pathogenic and likely pathogenic SVs as well as any SV around the genomic coordinates of the VOI (*USP53*) were examined and visualized with samplot. Of note, no SV in the VOI region was called for any of the samples. Five CNV regions of no interest (see next paragraph) but reported here for thoroughness were called as likely pathogenic or pathogenic by AnnotSV: (a) two pathogenic CNVs occurred in chromosome 4 but were present in all samples: a 304 bp deletion and a 117 kb deletion (chr4: 68508074-68625397); (b) chr19: 42813046-43241094 (pathogenic) was found in only one of the affected daughters; (c) two potentially pathogenic overlapping regions on chr20:29125607-30424527 and chr20: 29294818-30482698 were blacklisted (and excluded by Genvisis) and not examined further. A deletion (chr4:118712587-119795518; *USP53* region +/− 500 kb) was not called in the CNV set but was visualized.

Pedigree-based model: Rare & pathogenic (Appendix A). Because the two daughters are affected and the father is not, the vcf output from smoove was filtered additionally using the slivar tool for paternal heterozygosity and homozygosity in both affected daughters. Confirming the model-free analysis (see previous paragraph), only one CNV was found present in father and both affected daughters.

Pedigree-based model: Rare & unknown pathogenicity (Appendix A). The vcf filtered for paternal heterozygosity and homozygosity in affected daughters was searched with AnnotSV for deletions and duplications that were not annotated as 24 benign loss or benign gain, respectively. The outcome was a set of 39 CNV regions with unknown significance for pathogenicity. These CNVs were inspected using samplot. Five of the 39 CNVs were blacklisted, i.e., location in low complexity regions; incomplete reference genome assembly; bioinformatic misprocessing, or limitations inherent to cohort-specific private alleles (e.g., arising from genetic ancestries or sequencing kits) and were not considered further: chr10:132914133-132914719; chr13:113136636-113136949; chr7:58189288-61967062; chr7:60039029-60527184; chr7:60527462-6053519.

### 2.3. Structural Models

A structural model of USP53 was obtained from the AlphaFold Protein Structure Database (AF-Q70EK8; [21,22,23]). USP53 has not yet been studied using X-ray crystallography. DUET server predicted the missense mutations’ effects on structures which were then compared to each other using Mol* (https://molstar.org/, accessed on 24 August 2023) superposition (SFX). Views of USP53 Fingers with variants were generated with ICn3D (https://www.ncbi.nlm.nih.gov/Structure/icn3d/, accessed on 24 August 2023) showing the effects of the mutations at cysteine 228. Similar results were visualized with PDB’s Structure Comparison ‘Superpose’ (http://wishart.biology.ualberta.ca/SuperPose/, accessed on 24 August 2023) and Pairwise Structure Alignment (https://www.rcsb.org/docs/tools/pairwise-structure-alignment, accessed on 24 August 2023) with jPATCAT (rigid with default parameters) as well as with Dali server (http://ekhidna2.biocenter.helsinki.fi/dali/, Accessed on 20 October 2023).

### 2.4. Immunohistochemistry

All animal studies were performed under approved protocols by the IACUC. C57Bl/6J mice (either sex, age ~3 months) were anesthetized (i.e., ketamine/xylazine), and the chest was opened to enable perfusion via the left cardiac ventricle with cold phosphate buffered saline (PBS) containing 4% paraformaldehyde (PFA). Cold fixation (PBS/4%PFA) of extracted brain was performed for 24 h and followed with PBS containing 2% PFA/15% sucrose for 24 h. Subsequently, the brain was transferred to 30% sucrose (PBS) for 24 h. The brain’s ventral surface was embedded on a chuck using Tissue-Tek O.C.T. Compound (Sakura, Torrence, CA, USA) and was frozen in the cryostat at −21 °C. Transverse frozen sections were ~30 µm thick and were air-dried overnight onto positively charged glass slides then stored at −80 °C. A PAP pen (Abcam, Waltham, MA) was used to mark hydrophobic circles around each section to prevent bleeding across sections.

For immunofluorescent histochemistry, slice sections were permeabilized in PBS with 0.1% Triton X-100 (PBST) at room temperature (RT) for 10 min; blocked with 1% goat serum in PBS at RT for 30 min; overlaid with 100 μL primary antibody after removing excess blocking buffer; and incubated overnight at 4 °C. The sections were washed thrice with 1XPBS, each for 5 min. They were incubated at RT after being overlaid with 100 μL of secondary antibody. The concentrations used for IF of both primary and secondary antibodies were 1–2 μg/mL. A detailed list of biologicals, concentrations applied, and their characterization is shown in Appendix A.

Secondary antibodies (100 µL) included the following: (1) [Alexa Fluor 488]-conjugated goat anti-rabbit IgG, cross-absorbed; Thermo Fisher Scientific, Waltham, MA; A32731; diluted to 2 µg/mL in PBST, 1% goat serum, 1% BSA); (2) [Alexa Fluor Plus 594]-conjugated goat anti-mouse IgG (H+L), cross-absorbed; Invitrogen/ThermoFisher# A32742; diluted to 2 µg/mL in PBST, 1% goat serum and 1% BSA.

Control immunostaining employed the following primary stains: (1) no primary antibody; (2) rabbit polyclonal anti-keyhole limpet hemocyanin affinity purified by Bethyl Labs A150-104A-7 (Hamburg, Germany); 1 mg/mL diluted 1:200–1000 in in PBST, 1% goat serum, 1% BSA; and (3) antigen-absorbed anti-USP53 (above) with 10x excess human antigen APrEST78787 (beginning at residue 493; SRASAQIISSSKSQILAPGEK-ITGKVKSDNGTGYDTDSSQDSRDRGNSCDSSSKSRNR) in PBST, 1% goat serum and 1% BSA; (Atlas Antibodies, Stockholm Sweden).

Slides for anatomical identification were stained with crystal violet acetate (Nissl) and mounted with 10 µL Fluoromount-G (Invitrogen, Waltham, MA) and cover slip. Sections were visualized with Evos FL Auto Fluoresence/Light microscope (Thermo Fisher Scientific) with 4X (NA 0.13), 10X (NA 0.4), 20X (NA 0.8), and 40X (0.95) plan-apochromat objectives.

### 2.5. Immunoprecipitation and Western Blots

All procedures were done on ice and at 4 °C. The dissected brain (~500 mg) was added to 0.5 mL cold Syn-PER™ Synaptic Protein Extraction Reagent (SSPER, Thermo Fisher Scientific). Homogenization was performed with 10 passes using a 1 mL Dounce tissue grinder (Wheaton, Millville, NJ, USA) with a tight pestle. After centrifugation at 1200× *g* for 10 min, the pellet (P1; i.e., debris containing nuclei, mitochondria, etc., and noted to contain minimal USP53) was discarded. The supernatant (S1) was centrifuged at 15,000× *g* for 20 min at 4 °C. The cytosolic fraction (S2) was discarded, while the pellet (P2) containing synaptosomes was resuspended in 200 µL SSPER and 100 µL co-IP buffer [150 mM NaCl, 10 mm Tris-HCl pH 7.4, 1 mM EDTA, 1 mM EGTA, 1x HALT inhibitor cocktail (ThermoFisher), Triton X-100 1%, NP-40 0.5%].

The remaining procedures followed the Pierce protocol for extraction of immune complexes with 25 µL Protein A magnetic beads (Pierce; Thermo Fisher Scientific) except the amount of primary antibody was 2.88 µg. The beads were resuspended in 30 µL 3X Laemmli buffer before separation on SDS-PAGE and subsequent transfer to a PVDF membrane.

The membrane was blocked after transfer in blocking buffer (5% BSA in 1X TBST) for 60 min at RT with rocking. Subsequently, the membrane was incubated with the primary antibody (Anti-USP53 1:750 dilution; Anti-GRIP2 1:1000 dilution; see above) in 1X TBST for 60 min at RT followed by four subsequent washes with 1X TBST each for 5 min. The secondary antibody (goat anti-rabbit IgG, peroxidase conjugated 1:500 in 1X TBST) to probe the blot was added for 60 min at RT with rocking. Subsequently, the membrane was washed four times with TBST by rocking for 5 min each. The bands were detected using chemiluminescence with SuperSignalTM West Dura (Thermo Fisher Scientific, Waltham, MA, USA) and recorded using a Li-Cor (Lincoln, Nebraska, USA) system.

## 3. Results

### 3.1. Pedigree PSY01

A consanguineous pedigree, PSY01 (Figure 1A), was identified from the countryside outskirts of Lahore, Pakistan. The parents, who were first cousins, had seven children including two daughters (IV:3; IV:5) affected with psychosis during their teenage years (DSM-V schizophrenia). One infant daughter was deceased previously. The parents and the other four children were evaluated with structured interviews and judged to be free of psychiatric or neurologic disorders. There was no history of severe nutritional deficiencies, birth complications, drug abuse, delayed developmental milestones, or intellectual disability. None had apparent manifestation of a hearing loss or history of infantile jaundice (see below).

### 3.2. Case Summaries

#### 3.2.1. Proband IV:3

Proband IV:3 developed psychiatric symptoms around age 18 years with progressively severe symptoms and a declining functional status. She was interviewed at age 24 years. She completed five years of schooling. She frequently talked to herself. Screening by the mMINI [24] indicated a DSM-V diagnosis of schizophrenia without other comorbidities.

The psychotic symptoms included auditory hallucinations and delusions. Examples of psychotic symptoms were: (1) She appeared to have auditory hallucinations responding at times to unseen stimuli and could not explain the rationale behind her behaviors. (2) She exhibited disorganized thinking and was sometimes incoherent in her speech. (3) She displayed formal thought disorder as her answers to many questions were not logical or meaningful. She was treated previously with olanzapine, carbamazepine, and haloperidol; these medicines were discontinued over a two-month period in 2017 because of the development of a rash and a high-grade fever during hospitalization. Subsequently, clonazepam, lorazepam, aripiprazole, and sodium valproate were tried. However, her responsiveness and compliance were inconsistent during the months preceding her current assessment. Marriage was prearranged by the family at age 16 years. She had no children and remained unemployed. Of the two affected sisters, this proband had the more severe phenotype.

#### 3.2.2. Proband IV:5

Proband IV:5 developed psychiatric symptoms before age 15 years and was interviewed initially at 18 years of age. Her illness took a progressive course with a decline in functional status never permitting employment. She completed four years of schooling. Screening by the mMINI indicated a diagnosis of schizophrenia without other comorbidities. She frequently talked to herself. The psychotic symptoms included auditory hallucinations as indicated by her responding to unknown stimuli. She demonstrated paranoia requiring the presence of a family member to enable enough comfort for interview. Marriage was prearranged by the family at age 16 years. She had no children. She had been treated with olanzapine, but her compliance was poor. No formal assessments outside the mMINI were possible given her inability to cooperate. Her GAF score at the time of the interview was 35.

### 3.3. Variant Identification

#### 3.3.1. Exome Sequencing Identifies USP53 as a Sole VOI

Typical parameters in WES are shown in Appendix A; primer sequences used for Sanger confirmation are shown in Appendix A. Only seven rare homozygous variants surfaced in the affected individuals and heterozygous in the parent (Appendix A). Out of these, only the *USP53* variant was predicted as likely deleterious. AgileVCFMapper revealed three chromosomal regions that were homozygous in both affected individuals and heterozygous in their unaffected father (Figure 1). The regions mapped to chromosomes 4, 12, and 15 (Appendix A); however, the exome data did not reveal a likely pathogenic variant in the latter two regions.

Only the 14.7 Mb homozygous region on chromosome 4 (4q26) contained an exonic variant that affected an amino acid conserved in evolution across all vertebrate species examined (Figure 2B). This variant was predicted to be likely pathogenic by all six software packages used. This VOI with a high GERP score of 6.06 was identified in the ubiquitin specific protease 53 gene: *USP53*; NM_019050.2, NP_061923.2 (isoform 1, transcript variant 2); Chr4 (hg38): g.119,212,713-119,295,518; + strand; chr4 (hg38):g.119,260,513T>C; c. 682T>C; p. (Cys228Arg); Figure 2). Variants in all other genes were predicted benign or affected amino acids which were not conserved in evolution.

Sanger sequencing confirmed the variant was transmitted as expected for a Mendelian recessive disorder: two obligate carriers were heterozygous, whereas one unaffected offspring lacked the variant (Figure 1C). The variant was absent from the DNA of 400 unrelated ethnically matched individuals and was thus not a polymorphism.

#### 3.3.2. Genome Sequencing Did Not Yield Additional Pathogenic SNVs

WES can suffer from limitations related to exon capture and to detection of structural variants. Therefore, WGS was performed after WES for the two affected girls and their father. The organization of WGS results follows the GATK pipeline (Appendix A). *USP53* c.682T>C, p.(Cys228Arg) was the sole SNV under the category “VOI/Previously Reported in the Literature” (Appendix A) converging with the results of the WES above (Appendix A). No variant in this category had LoF. Genome sequencing yielded additional variants under different inheritance models as categorized in software for processing the whole genome (Appendix A). No SNVs fit the genetic models (see Section 2.2).

#### 3.3.3. Genome Sequencing Did Not Yield a Likely Pathogenic CNV

No copy number variants (CNVs) were called in the region around *USP53* for any of the samples. Five CNV regions in the genome were called as likely pathogenic or pathogenic by AnnotSV. Two CNVs were excluded as they were present in all three samples (both affected daughters and unaffected father) and mapped to chromosome 4 (Appendix A). One 2.5 kb deletion was detected on chromosome 5, but it occurred only in the father. A 427 kb deletion located on chromosome 9 was found only in one of the affected daughters.

#### 3.3.4. Computational Modeling of USP53 Variants

Figure 3 shows the predicted structure of USP53 variants (both *mambo* and PSY01) visualized based on their nucleotide sequence and AlphaFold (model AF-Q70EK8-F1). The mutation occurs in the distal Fingers region near the Zn^2+^ binding motifs. However, whether USP53 binds Zn^2+^ is unknown.

DUET predicts the mutations’ effects (Figure 3B). Three structures (C228, C228R, C228S) were compared to each other using Mol* superposition (SFX). There appear minimal effects on the catalytic domains’ overall globular structure. However, the C228R variant likely affects binding to ubiquitin chains and may affect binding of monoubiquitin. Similar structural results were visualized with PDB’s Structure Comparison “Superpose” and Pairwise Structural Alignment with jPATCAT (rigid with default parameters) as well as with Dali.

#### 3.3.5. USP53 Is Expressed in the Mouse Hippocampus

The antibody specificity was established through three separate controls: (1) no primary antibody; (2) ten-fold excess USP53 antigen (APrEST78787; Sigma Aldrich)) blocking anti-USP53; and (3) anti-KLH (an antigen foreign to mice). Reagents, sources, and characterization are shown in Appendix A.

Immunohistochemistry localized USP53 mostly to the olfactory bulb, cerebellum, and hippocampus. Localization of USP53 throughout mouse brain will be detailed elsewhere. Here, the principal focus is on the hippocampus because of its relevance to schizophrenia [25,26,27,28]. Hippocampal regions CA 1–3 and the granular dentate were stained densely by USP53, GRIA2, and GRIP2 antibodies (Figure 4). In some neurons, GRIP2 and GRIA2 localized toward cell margins (Figure 4G,H). The sections were not identical; so, the question of cellular and subcellular localization of these proteins will require higher resolution studies. However, preliminary studies suggest USP53 colocalizes with PSD95 at the spines in cultured rat hippocampal neurons (unpublished observations).

#### 3.3.6. Anti-USP53 Pulls-Down GRIP2 and GRIA2 (GluA2) from Murine Synaptosomes

Western blots (WBs) with the primary antibodies probing synaptosomes from the lysate of whole murine brain identified appropriate molecular weight species for USP53, GRIP2, and GRIA2 (Figure 5A–C). The pull-down studies are shown in Figure 5D–L and demonstrate co-immunoprecipitation of all pairs of these three proteins. For example, anti-USP53 specifically immunoprecipitates a protein of molecular weight consistent with USP53 (~130 kDa) from murine synaptosomes (Figure 5A). Protein A magnetic beads pull down complexes of anti-USP53 incubated with antigens from synaptosomes (Figure 5D). After washing, the complexes were eluted from the beads. The western blot again showed immunoprecipitation of USP53. Note that a band of molecular weight ~50 KDa was considered related to the large antibody load from Protein A beads and the limited specificity of the peroxidase conjugated secondary antibody. It is seen typically in immunoaffinity purification with Protein A. Figure 5M,N are controls. Again the ~50 KDa band appears when using anti-keyhole limpet hemocyanin (foreign to mice) and Protein A beads, but it is absent when no antibodies were used along with Protein A magnetic beads (compare M and N).

## 4. Discussion

The current study presents a multiplex consanguineous family affected by an extreme phenotype: early onset (teenage years), severe (disabling), treatment-resistant psychosis. The phenotype enabled ascertainment because all members of the pedigree provided clear classification as to the presence or absence of psychosis and the absence of psychiatric comorbidity. The target phenotype was based on symptoms and was independent of clinical DSM-V [30] or theoretical RDoC [31] labels given ongoing concerns about psychiatric nosology. Nevertheless, all psychotic participants met DSM-V criteria for schizophrenia that is widely considered a neurodevelopmental disorder.

Application of both exome and genome sequencing, homozygosity mapping, and filtering produced only one rare VOI. Searching public databases (gnomAD v.2.1) found seven heterozygous USP53 p.C228R alleles out of 280,128; MAF 2.5(10)^−5^. The “control” gnomAD v2.1.2 dataset listed four missense heterozygous USP53 p.C228R alleles out of 119,204; MAF 3.3(10)^−5^. In the gnomAD 3.1.2 dataset, one homozygous missense variant of USP53 p.C228R occurred (13 alleles out of 152,1721; MAF 8.54(10)^−5^). The interpretation of this homozygote is ambiguous because the definition of “normal” was with respect to the disease studied in the GWAS rather than to an absolute sense (i.e., without formal psychiatric assessment of lifetime presence of diagnosis). So, whether the homozyote had a documented lifetime history of psychosis or schizophrenia is unknown.

Discovery of the *USP53* variant was difficult to contextualize. USP53 had limited functional data without published relevance to mental illness. The protein had not been identified in the CNS. This lack of information precluded interpretation of the neuroscience context and prompted initial functional studies of USP53.

The staining of USP53, GRIP2, and GRIA2 in the CA fields and the granular dentate suggests involvement of these medial temporal structures in the pathophysiology of schizophrenia and psychosis consistent with broad literature (see above). Indeed, the latest PGC meta-analysis points to the cells in these regions as having the highest probability for association with enrichment of gene products from the schizophrenia GWAS [32]. The co-immunoprecipitation of USP53 with GRIP2 and GRIA2 indicates USP53 is likely involved in the AMPA synapse, but the precise pathological mechanism of the identified variant remains to be discovered (such as via involvement of additional receptors including NMDA-R, metabotropic glutamate receptors, GABA, muscarinic, etc.).

Complex disorders such as psychiatric diseases are thought to be mostly polygenic with some degree of contribution from environmental factors. Mendelian inheritance is comparatively unknown for common psychiatric disorders such as schizophrenia [3]. However, a few syndromes due to single gene variants have psychosis as an accompanying feature which suggests that some psychiatric disorders could be attributable to or approach Mendelian forms of genetic defects.

Previously, a locus for schizophrenia with comorbidities was mapped to chromosome 22 in a Pakistani family with three affected daughters born to unaffected consanguineous parents [33]. Also, ceroid lipofuscinosis (neuronal 8, Northern epilepsy variant, OMIM #610003) is caused by biallelic variants of *CLN8*. These patients can exhibit behavioral psychiatric problems including irritability, attention deficits, and restlessness with onset in adolescence. Similarly, patients with Wolfram syndrome (OMIM # 222300) frequently exhibit psychiatric symptoms due to biallelic *WFS1* variants. Heterozygous variants have also been reported in inherited psychiatric disorders. Monoallelic rare coding variants in *GRM5*, *PPEF2* or *LRP1B*—all related to NMDA receptor (NMDAR; mGlu5) biology—were found in some affected participants in five multiplex nuclear families with schizophrenia [32]; the variants’ absence in unaffected family members was not reported In addition, rare de novo variants in single genes were correlated with psychiatric disorders by many groups [34,35].

### 4.1. USP53 Structure and Function

No prior data addressed the involvement of USP53 in the CNS or in glutamatergic neurotransmission, psychosis, or any mental disorder. The only pertinent data showed widespread low-level transcription of *Usp53* in the murine brain with the highest density in the olfactory bulb (Allen atlas). USP53 transcripts were also located in several other organs such as skeletal muscle, heart, and testis.

The immunohistochemistry experiments found USP53 expression in the murine olfactory bulb, hippocampal CA fields and dentate, and the cerebellar cortex. All are structures acknowledged to play major roles in neuroplasticity, learning, and memory. These regions also have dense GRIA2 distribution [36].

USP53 is a multi-domain protein (Figure 2A). This USP53 variant affects an amino acid that is highly conserved across all vertebrates (Figure 2B). USP53 has no enzymatic activity because a histidine in its catalytic triad is missing. The lack of enzymatic activity was previously confirmed experimentally [37,38].

### 4.2. Murine USP53 Cys228Ser (mambo) Variant

A murine *Usp53* mutant (*mambo, mbo*) was reported previously with a serine substitution at Cys228 (GRCm39; c.123,751,372), [38] precisely where the human variant associated with psychosis has an arginine substitution. This mouse is not available for study; the only reported study was auditory. *Mambo* was assessed to have progressive hearing loss. The long C-terminal tail of USP53 interacted with TJP1 and TJP2 potentially impairing barrier function with lowered endocochlear potential and consequent hearing loss. Of interest, many synaptic proteins are shared with other cytoskeletal specializations (occludin, vinculin, α-actinin). Incidentally, we have found USP53 also localizes outside the CNS at focal adhesion plaques in HEK293 cells transformed with USP53^+^ constructs (unpublished observations).

### 4.3. Human USP53 Pathogenic Variants

Twenty-eight *USP53* pathogenic variants have been found associated with different phenotypes (Human Gene Mutation Database HGMD; accessed October, 2023) (Table 1). A patient with the variant *USP53* c.951delT; p.Phe317fs was found segregating the phenotype of speech and developmental delay along with itching and benign recurrent intrahepatic cholestasis [39]. In another instance, a heterozygous SNV *USP53*, chr4:120190897, c.1340A>G, p.(Glu447Gly)was one among 17 other candidates identified in a multiplex family with nine individuals suffering from bipolar disorder [40]. A heterozygous *USP53* frameshift variant 4:120190846:CTAAGT:C was identified in a study of de novo variation in 97 trios of Ashkenazi Jewish descent with early onset bipolar disorder and without a family history of bipolar disorder (“simplex”) [41]. Whether these bipolar patients had a history of psychosis was not detailed. No *USP53* variants have been previously reported specifically associated with schizophrenia.

In summary, studies of humans with *USP53* variants report variable phenotypes with predominance of cholestasis especially when missing the C-terminal tail that binds to tight junctions of hepatocytes and cochlear cells (Table 1). Based on the findings above, a *USP53*-related phenotype was added to OMIM (OMIM# 619658).

### 4.4. Ubiquitin, USP53 Interactome, and the AMPAR Synapse

Deubiquitinating enzymes (DUBs) typically include domains both for catalytic removal of ubiquitin and for protein-to-protein interactions (PPIs) thereby interacting not only with ubiquitin but also ubiquitin processing enzymes. The ubiquitin system has an important function in AMPAR processing. For example, E3 ligases modulating AMPAR ubiquitination include NEDD4-1 [42]; APCCdh1 [43]; and RNF167 [44].

Two catalytic DUBs, USP8 and USP46, participate in AMPA-R deubiquitination [45]. The ubiquitin code is essential for intracellular trafficking of AMPA receptors (AMPAR) and coupling to neuronal activity [46]. For USPs without protease activity such as USP53, function is probably critically intertwined not only with ubiquitin and ubiquitin adducts but also with other proteins likely involved in cell signaling, neurotransmission, scaffolds, and other related processes. Of note, many cytoskeletal and docking proteins participating in classical cell-to-cell interactions appear redeployed at the synapse. These interactions are frequently dynamic and transient. Therefore, defining the USP53 interactome must be a priority to understand the biology of USP53 as related to psychosis.

A prominent theory about schizophrenia and psychosis involves glutamatergic dysfunction. Although initially focused on NMDA-R hypofunction, the theories have expanded to include AMPA-R and mGlu-R. Glutamate theories derive from wide converging support from multiple sources: GWAS risk alleles, cited previously; GluR autoimmune encephalitis [47]; NMDA antagonists (ketamine or PCP) [48]; metabotropic GluR [49]; and GluR post-translational modifications [50]. Because NMDA-R, AMPA-R, and mGlu-R interact in complex ways [51,52], multiple glutamate receptors are likely involved.

Since there was little prior information about USP53 related to neurodevelopment or psychiatric disease, preliminary immunolocalization and immunoprecipitation studies were conducted to determine whether the variant was at all consistent with the above hypothesis. Our experiments localized USP53 most intensely to major regions involved in neuroplasticity, learning, and memory (olfactory bulb, hippocampus, and cerebellum). Given the postulated fundamental role of the hippocampus in the pathology of schizophrenia (see above), the medial temporal lobe was examined further with respect to USP53.

The staining in similar regions as well as the co-immunoprecipitation of GRIA2 and GRIP2 with USP53 point to the AMPA synapse as a potential central site of pathophysiology. The present findings compel the need for precise dissection of the interactions of WT and mutant USP53 with other synaptic proteins associated with the processing of AMPA-R as targets for a potential etiological mechanism.

These data directly link a key etiological hypothesis about psychosis/schizophrenia to a variant inherited in a Mendelian recessive mode in a consanguineous family. Alternatively, prior research in schizophrenia also indicated another hypothesis related to dysfunction in the ubiquitin system per se [for recent review, see [53]. Although there is an evolving literature on ubiquitin processing in glutamatergic neurotransmission [54], involvement of USP53 has not yet been implicated.

**Table 1 genes-14-01921-t001:** Previously reported *USP53* variants and associated phenotypes.

Ethnicity/ Origin	Onset Age ‡	Gender	Consanguinity	Variant	USP53 Transcript	Zygosity	Variant Type	Phenotype ± Comorbidities	Reference
Pakistani	16 years	F	Yes	c.682T>C; p.Cys228Arg	NM_019050.2	Homozygous	Missense	Psychosis w/o cholestasis	Current
Syrian	4 months	F	Yes	c.238-1G>C	NM_019050.2	Homozygous	Splice site	Cholestasis	[55]
Brazilian	10 days	M	No	c.1687_1688delinsC p.Ser563Profs * 25	NM_019050.2	Homozygous	Frameshift	Cholestasis + Jaundice	[56]
Saudi Arabian	N/A	F	Yes	c.951delT p.[Phe317Leufs*6]	NM_019050.2	Homozygous	Frameshift	Hypocalcemia itching, deafness w/o cholestasis	[57]
Ashkenazi	15 years^‡^	F	NA *	c.1295_1299delTAAGT p.Leu432fs	NA *	Heterozygous	Frameshift	Bipolar I	[41]
Chinese	3 days	F	No	c.1012C>T p.Arg338Ter	NM_019050.2	Homozygous	Nonsense	Jaundice	[58]
Chinese	2 days	M	No	c.169C>T/c.831_832insAG p.Arg57Ter/p.Val279GlufsTer16	NM_019050.2	Compound heterozygous	Frameshift	Jaundice	[58]
Chinese	6 months	F	No	c.569+2T>C/c.878G>T p:?/p.Gly293Val	NM_019050.2	Compound heterozygous	Splice/Mis-sense	Jaundice	[58]
Chinese	5 months	M	No	c.581delA/c.1012C>T p.Arg195GlufsTer38/p.Arg338Ter	NM_019050.2	Compound heterozygous	Frameshift/Nonsense	Jaundice	[58]
Chinese	1 month	F	No	c.1012C>T/c.1426C>T p.Arg338Ter/p.Arg476Ter	NM_019050.2	Compound heterozygous	Nonsense	Jaundice	[58]
Chinese	5 months	M	No	c.395A>G/c.1558C>T p.His132Arg/p.Arg520Ter	NM_019050.2	Compound heterozygous	Missense/Nonsense	Jaundice + hearing loss	[58]
Chinese	7 months	M	No	c.297G>Tc.1012C>T p.Arg99Ser/p.Arg338Ter	NM_019050.2	Compound heterozygous	Missense/Nonsense	Jaundice	[58]
Turkish	3 months	F	No	Deletion of 1st coding exon	NM_01050.2	Homozygous	Gross del	Cholestasis	[59]
Turkish	2 months	M	No	Deletion of 1st coding exon	NM_019050.2	Homozygous	Gross del	Cholestasis with heart failure	[59]
Middle Eastern	5 months	F	No	c.145–11_167del	NM_019050.2	Homozygous	Gross del	Cholestasis	[59]
Middle Eastern	7 years	M	No	c.145–11_167del	NM_019050.2	Homozygous	Gross del	Cholestasis	[59]
North African	15 years	M	No	c.725C>T p.Pro242Leu	NM_019050.2	Homozygous	Missense	Cholestasis	[59]
South Asian	4 years	M	No	c.510delA p.Ser171ArgfsTer62	NM_019050.2	Homozygous	Frameshift	Cholestasis	[59]
Japanese	16 years	F	Yes	c.1744C>T p. Arg582 *	NM_019050.2	Homozygous	Nonsense	Benign recurrent intrahepatic cholestasis + itching + hypothyroid	[39]
Dagestan/ Russian	Neonatal	M	Yes	c.1017_1057del p.Cys339Trpfs	NM_019050.2	Homozygous	Frameshift	Prolonged jaundice	[60]
Indian	6 months	M	Yes	c.822+1delG	NM_019050.2	Homozygous	Splice site	Jaundice with pruritus	[61]
Pakistani	8 months	M	Yes	c.1524T>G p.Tyr508 *	NM_019050.2	Homozygous	Nonsense	Jaundice + thrombocytosis+ itching + pigmented stools	[62]
Pakistani	NA *	NA *	NA *	c.169C>T p.Arg57 *	NM_019050.2	Homozygous	Nonsense	Early-onset intrahepatic cholestasis	[62]
Pakistani	NA *	NA *	NA *	c.475_476delCT p.Leu159fs	NM_019050.2	Homozygous	Frameshift	Early-onset intrahepatic cholestasis	[62]
Pakistani	NA *	NA *	NA *	c.822+1delG	NM_019050.2	Homozygous	Splice site	Early-onset intrahepatic cholestasis	[62]
Pakistani	NA *	NA *	NA *	c.1214dupA p.Asn405fs	NM_019050.2	Homozygous	Frameshift	Early-onset intrahepatic cholestasis	[62]
American	NA *	F	NA *	Duplication of 375 kb incl. entire gene + MYOZ2 + FABP2 (described at genomic DNA level)	?	?	Gross Ins/Dup	Cantu syndrome with coarse facial features	[63]
French–Canadian	NA *	NA *	NA *	75 kb partial gene + FABP2 + C4orf3 (described at genomic DNA level)	?	?	Gross del	Confocal epilepsy	[64]

‡ average age of group; * not available (NA); ?, unknown.

### 4.5. Limitations

The identification of disease-associated, rare, Mendelian recessive variants based on the approach used here is well-established in medical genetics [65,66]. However, this approach has seen limited application in finding Mendelian genes for common mental illness [3,17,18]. Given the MAF and consanguinity suggesting a rare genetic form of schizophrenia, the variant will not be a common cause despite the common phenotype. Evidence for the variant’s possible pathogenicity was based on in silico analyses and evolutionary conservation. Future work will require site-directed mutagenesis of USP53 and additional functional studies for definitive conclusions. Also, VOIs cannot provide direct evidence for pathogenicity or causality, and thus interpretation requires caution. Indirect evidence for pathogenicity in medical genetics can come from aggregating cases with similar phenotypes associated with the same mutated allele or gene—exceedingly unlikely for ultra-rare variants or for diseases with high genetic heterogeneity. Alternatively, disease biomarkers can be associated with the genotype which common psychiatric disorders lack requiring reliance only on observed or reported behaviors for diagnosis. Functional studies will thus require triangulation between electrophysiological, molecular, and cellular approaches. The latter was begun here.

## 5. Conclusions

We identified a novel missense variant in *USP53* associated with autosomal recessive transmission of severe psychosis/schizophrenia in a multiplex consanguineous family. The variant was selected based on WES/WGS, homozygosity mapping, in silico evidence of damaged protein, rarity, evolutionary conservation, and absence in databases of normal subjects. Since no data evidenced prior expression of USP53 in the CNS or involvement in any psychiatric disorder, functional studies were performed. USP53 surfaced in the same brain regions (hippocampus, dentate) as GRIA2 and GRIP2. All three proteins appear components of the AMPA synaptic interactome based pull-down studies using co-immunoprecipitation. If confirmed, these data provide the first Mendelian gene for a common psychiatric phenotype that is a rare genetic disorder. Such genes provide historically new routes for foundational advances in translating pathophysiology to clinical practice.

## Figures and Tables

**Figure 1 genes-14-01921-f001:**
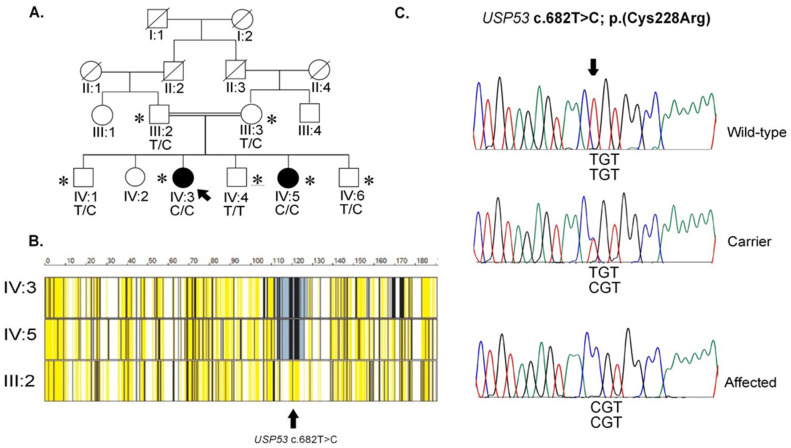
Pedigree, autozygous region, and Sanger sequence traces. (**A**). Family PSY01. Two first cousins had seven children of whom two (IV:3, IV:5) were diagnosed with psychosis (meeting DSM-V schizophrenia). Squares and circles depict males and females, respectively. Filled symbols denote affected individuals. Double line indicates the presence of consanguinity. The genotypes for the *USP53* variant are indicated below each participant’s symbol. (**B**). AgileVCFMapper output. The display shows chromosome 4q26, including the approximately 17 Mb autozygous region (from ~111 Mb-~128 Mb, coded in blue/grey color) that harbors the *USP53* c.668T>C (hg19); p.(Cys228Arg) variant. Color coding indicates shared homozygous regions as large blocks of blue/grey to black in the affected individuals’ data; the same being heterozygous in the parent, as indicated by the high number of bars in yellow. (**C**). Representative sequence traces. Electropherograms showing wildtype sequence in unaffected sibling (IV:4, Top panel); unaffected carriers (III:2, III:3, IV:1, IV:6. Middle panel); and affected individuals (IV:3, IV:5, Lower panel) for *USP53* c.668T>C variant. Color coding: red, T; blue, C; bright green, A; black, G. An asterisk (*) depicts the participating individuals of the family. Additional electropherograms are included in the Appendix A.

**Figure 2 genes-14-01921-f002:**
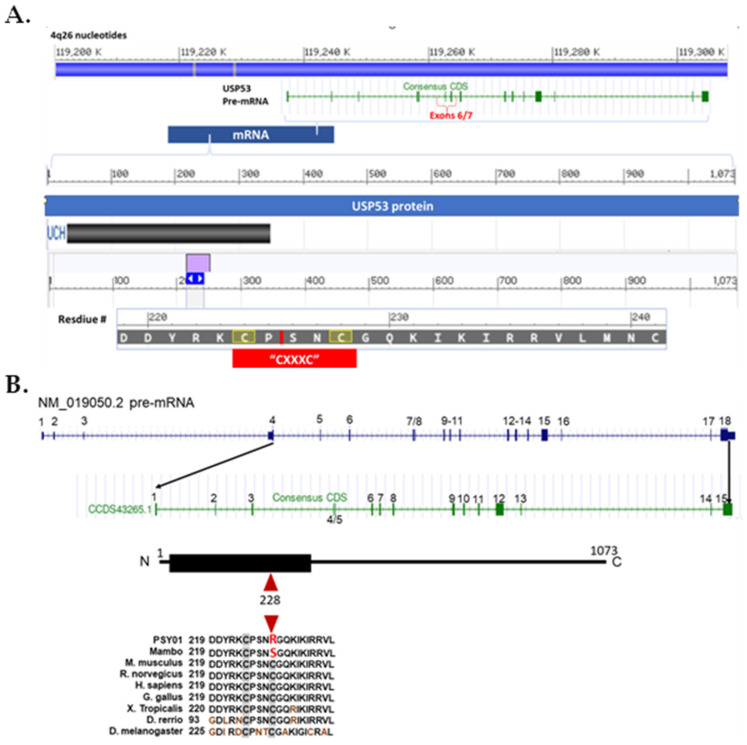
Human *USP53* gene, cDNA pre-mRNA, and protein. (**A**). Coding region (CCDS43265.1) of wild type human *USP53* within chromosome 4q26 (119,212,713-119,295,518; GRCh38/hg38) displayed with UCSC genome browser indicating the 18 exons of the gene (vertical green bars or rectangles; upper panel). The coding region begins at exon 4 of the pre-mRNA (NM_019050.3) and ends with exon 15 (upper panel). The middle panel shows the USP53 protein spanning amino acid residues 1-1073 with a catalytic domain (i.e., Ubiquitin C-terminal Hydrolase, UCH superfamily, amino acids 29-352) displayed as a black rectangle. The purple rectangle shows the region of interest in USP53 for the mutation that is shown in the lowest panel (amino acids 220-240). The small red vertical bar within the amino acid sequence (between P and S) shows the splice site between exons 6 and 7 of the pre-mRNA. The red rectangle in the lowest panel shows the Cys motif (“CXXXC”) that spans the Finger region of interest. The second cysteine (C228) in the red rectangle denotes the mutated cysteine in the patients substituted by arginine (and in *mambo* is mutated to serine; see (**B**)). After splicing, the *USP53*, NM_019050.3 mRNA codes for 1073 amino acids. The UCH catalytic domain is shown as a black rectangle. The amino acids from 220-240 are displayed in the black bar at the bottom of the figure. (**B**). Diagram of NM_019050.3 and relationship to CCDS43265.1. The red star identifies the seventh exon, the location of the mutation. The lowest panel schematic shows USP53 with conservation of amino acids across multiple vertebrate species. The mutation at residue 228 for both PSY01 and *mambo* is shown in red. Brown symbols denote residues that differ from the canonical sequence.

**Figure 3 genes-14-01921-f003:**
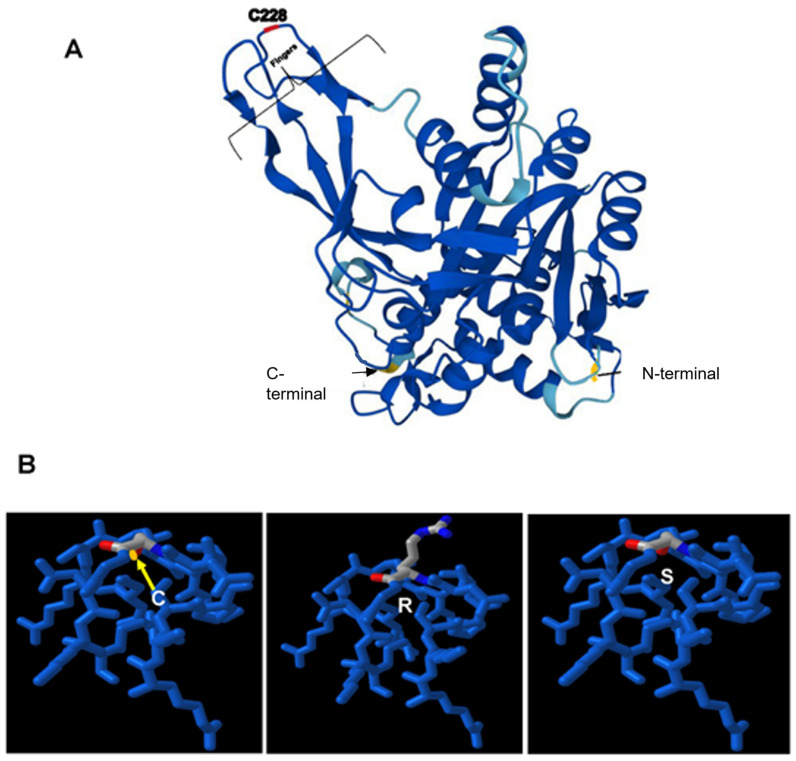
Structure of Mus musculus USP53 (P15975). (**A**). Model generated with AlphaFold (chain A; monomer; biological unit). The confidence in the modeled structure is parametrized in color (dark blue, very high >90%; light blue, confident 70–90%; yellow, low 50–70%). The terminal regions had low confidence, so they were mostly removed to enable visualization of the segments with greater confidence. The residues colored gold at the C-terminus contains amino acids H370, Y371 and R372, while that at the N-terminus is P28. The mutated amino acid associated with psychosis is shown in red (residue 228) localizing to the Fingers region. (**B**). Models of the surface of Fingers (Figure 3A, upper left) generated with ICn3D showing the mutation at C228 (left panel), R228 (middle panel; PSY01), and S228 (right panel; *mambo*). The overall globular structure of the catalytic unit appears little changed as the arginine mutation points into the solvent. However, it remains unclear whether arginine substitution in the Fingers region will result in steric hinderance of potential binding partners. Red atoms denote oxygen, dark blue (next to grey carbons) denotes nitrogen; yellow denotes sulfur (barely visible in left panel behind the alpha carbon in grey).

**Figure 4 genes-14-01921-f004:**
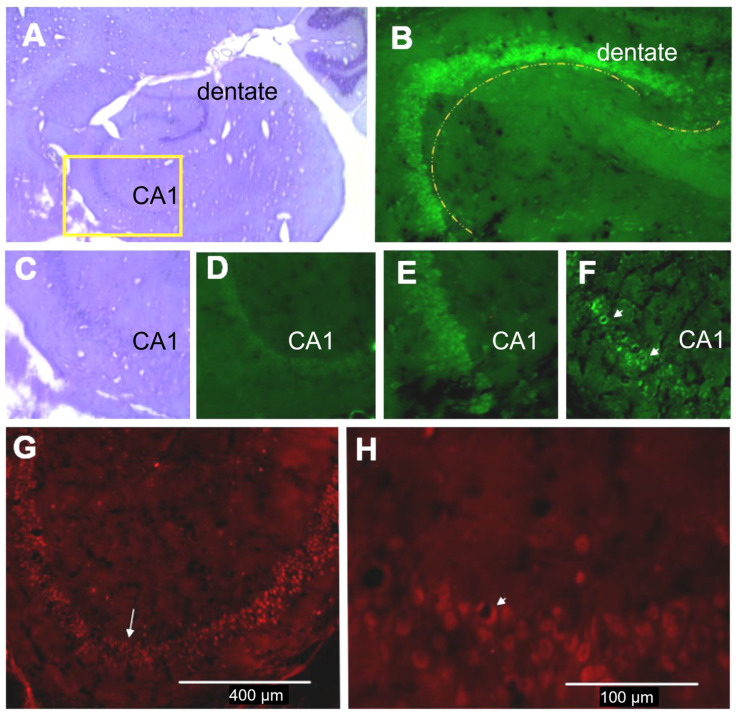
Immunofluorescent localization of USP53, GRIP2, and GRIA2 in murine hippocampus. (**A**). Nissl (neuronal) staining of hippocampal region, 4X; region in gold square is magnified in C, 20X. (**B**). USP53 antibody staining shows regions of CA1–3 and dentate (dashed arc: left, CA; right, dentate), 20X. (**C**). Nissl staining CA1, 40X. (**D**). Control (no primary antibody), 20X. (**E**). USP53 expression in CA1, 20X. (**F**). GRIP2 expression in CA1, 20X; note dense staining at what appears as a neuron margin (arrows). (**G**). GluA2 expression in CA1, 20X; note marginal staining (arrow); bar 400 μm. (**H**). GluA2 expression in CA1, 40X; note marginal staining (arrow); bar 100 μm.

**Figure 5 genes-14-01921-f005:**
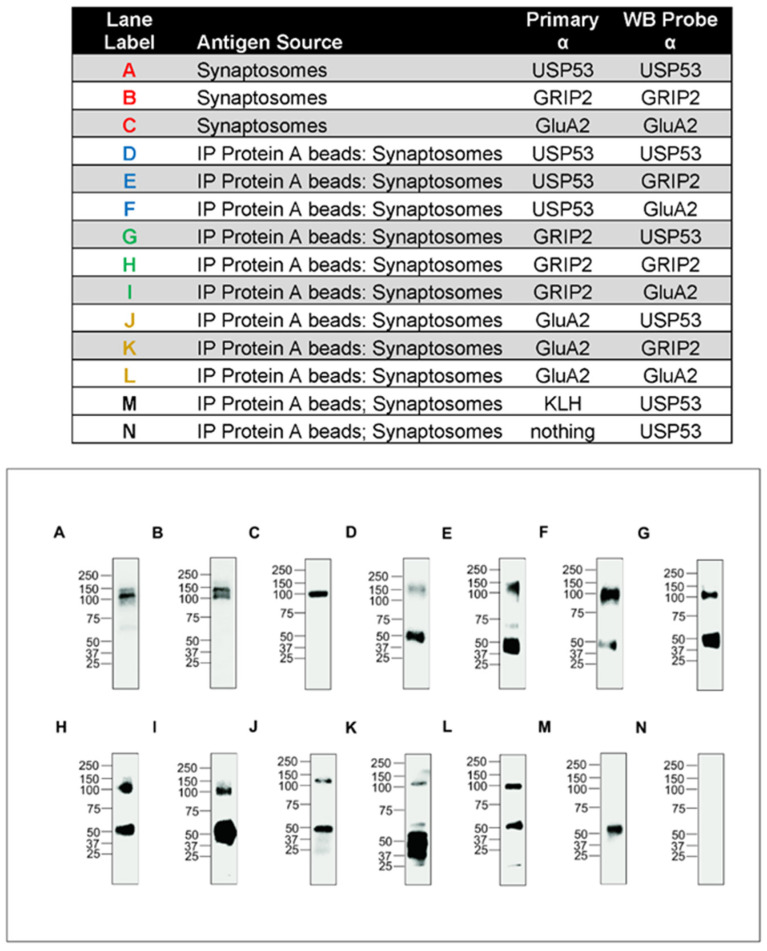
Co-immunoprecipitation of USP53 with GRIP2 and GluA2/GRIA2 from mouse brain. Lane labels in top panel summarize experimental conditions. The lanes are copied from several 8% SDS-PAGE gels obtained in different experiments therefore requiring individual weight markers. Note that each lane is separate and not compared to other lanes (i.e., not a “comparative blot). (**A**). Anti-USP53 identifies USP53 (~120 kDa) in synaptosomes (WB blot). (**B**). Anti-GRIP2 identifies GRIP2 (~130 kD) in synaptosomes (WB); the lower band is likely ABP, AMPA-receptor Binding Protein, a splice variant of lower molecular weight (~98 kD vs. 130 kD) [29]. (**C**). Anti-GluA2 identifies GluA2 (~100 kDa) in synaptosomes (WB). (**D**). IP of USP53 with anti-USP53 WB. (**E**). Co-IP of USP53 with anti-GRIP2 WB. (**F**). Co-IP USP53 and anti-GluA2 WB. (**G**). IP of GRIP2 with anti-USP53 WB. (**H**). Co-IP GRIP2 with anti-GRIP2 WB. (**I**). Co-IP GRIP2 with anti-GluA2 WB. (**J**). Co-IP GluA2 with anti-USP53 WB. (**K**). IP of GluA2 with anti-GRIP2 WB. (**L**). Co-IP GluA2 with anti-GluA2 WB. (**M**). Co-IP rabbit anti-KLH (negative control) with anti-USP53 WB. (**N**). Co-IP negative control (no primary antibody) with anti-USP53 WB. A band around ~50 kDa was present when using Protein A beads with primary antibody, but not if Protein A was not used (as in synaptosomes: A–C) or if primary antibody was omitted (N) [29].

## Data Availability

The datasets generated during and/or analyzed during the current study are not publicly available due to refusal by the PSY01 family for disclosure because of the sensitivity of the data (pedigree, personal medical information: psychiatric diagnoses, rating scales, potential for de-identification). These are available from SN on reasonable request.

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
