# Peer review of "Genome Sequencing of Consanguineous Family Implicates Ubiquitin-Specific Protease 53 (USP53) Variant in Psychosis/Schizophrenia: Wild-Type Expression in Murine Hippocampal CA 1–3 and Granular Dentate with AMPA Synapse Interactions"

_genes, 2023, doi:10.3390/genes14101921_

Round 1

Reviewer 1 Report

In this manuscript, Kanwal et al. identify a rare, predicted deleterious variant of USP53 in two affected siblings from a consanguineous family. This identification was accomplished through a rare variant analysis using both exome and genome sequencing data. In silico functional prediction, population data, and evolutionary conservation all suggest that this variant could have deleterious effects. I agree with the authors in stating that presenting this family's genetic data would be a valuable contribution to the scientific community.

However, I do have one major comment concerning the genomic analysis performed on the case. Since genomic sequencing data for the family are available but not open for sharing due to consent reasons, it becomes particularly important to clearly describe the analysis that was conducted. Some additional analysis should be considered based on the current genome analysis pipeline.

Both SNV and small insertion or deletions (indels) can be robustly accessed with WES and WGS pipeline, Can the author confirm rare indel has been accessed with WES and WGS? If that is the case, it would be good to add that in the method description. Also since WGS data is available, it would be good to perform both copy number imbalance (CNV) and balanced structural variant (SV) in the analysis, is there any balanced translocation, inversation identified?

The author needs to define what is the gene damage index (GDI). Does this include a list of candidate genes? If that is the case, the list of genes need to be included.

Can the author confirm the first cousin consanguinity with WES or WGS data, e.g., what is the total size of Runs of homozygosity (ROH) and what is the inbreeding coefficient? The size of total ROH could be estimated through calculating the size of  absence of heterozygosity(AOH) region.

There are ~ 20 genes that have been associated with schizophrenia in omim, can the author confirm any rare pathogenic variant, SNV, indel, CNV, SV calls around these locus?

In addition, I have some minor comments on the figure, figure legend and text to better represent the data:

For Fig1, the figure legend of 1B does not explain how the ROH region was shown in the plot. The sanger traces of each individual should be displayed.

In the supplementary SNV discovery section, what is the IGV manual inspection criteria for good, possible, and bad?

4.4, paragraph 4, ‘neurodevelopmental disease’ might be more specific than ‘neuroscience’

‘Typical parameters in WES are shown in ST1’, the ST1 does not seem to have the parameters.

Section 2.3.1, paragraph 3, ‘... based on examination of 800 chromosomes.’ Does the author mean 800 individual exome sequencing data/ genome? If this is an internal database, the experiment to construct the database needs to be clarified.

Section 2.3.2 paragraph 1, ‘No SNVs categorized as “good” fit the genetic models.’ the author needs to define what is a good fit and the genetic model tested.

Author Response

We thank the three reviewers for the suggestions to improve the work. Revisions are marked with red in the original ms (MARK UP) because there were so many changes that reading became difficult..

REPLY TO REVIEWER SUGGESTIONS

Reviewer #1

a) …” clearly describe the analysis that was conducted….”
We have greatly expanded the section on Methods to clarify the analysis. Also included are the SVs (indels, CNVs) found and the rationale for dismissing them as VOIs. We now include a diagram (Supplementary Figure 1) outlining the approach (based on GATK). 

  1. b) Both SNV and small insertion or deletions (indels) can be robustly accessed with WES and WGS pipeline, Can the author confirm rare indel has been accessed with WES and WGS? If that is the case, it would be good to add that in the method description. Also since WGS data is available, it would be good to perform both copy number imbalance (CNV) and balanced structural variant (SV) in the analysis, is there any balanced translocation, inversation identified?”
    Please see above. We clarify that WGS data were derived from the father and the two affected daughters (i.e., not all pedigree members genome sequencing). Therefore, normalization methods were not possible (GATK recommends a minimum of N = 30). WGS was used to look for SNVs, CNVs, and other SVs. No balanced translocations or inversions were identified. A diagram (Supplementary Figure 1) outlines the calling pipeline. Rare indels were accessed with WGS and are noted in the methods None were of interest to this pedigree. For example, “One 2.5 kb deletion was detected on chromosome 5, but it occurred only in the father. A 427 kb deletion located on…”
  2. c) The author needs to define what is the gene damage index (GDI).
    “To minimize the risk of calling false positives, genes that have undergone a heavy mutational load in the population have been tabulated {Itan, 2015 #18418}. The gene damage index (GDI) was used as a metric of the accumulated mutational damage of each human gene in the healthy human population based on the 1000 Genomes Project database. These data were merged into the VOIs in Table 4-6.” We have merely added them to our tables.
  3. d) Can the author confirm the first cousin consanguinity with WES or WGS data, e.g., what is the total size of Runs of homozygosity (ROH) and what is the inbreeding coefficient? The size of total ROH could be estimated through calculating the size of absence of heterozygosity(AOH) region.
    Our data are consistent with consanguinity given the 3 large runs of homozygosity (e.g., 14.7 Mb). However, given the short amount of time (10 days) permitted for revisions, we are unable to get the data from archives, measure ROHs, and estimate inbreeding coefficient.
  4. e) There are ~ 20 genes that have been associated with schizophrenia in omim, can the author confirm any rare pathogenic variant, SNV, indel, CNV, SV calls around these loci?
    There are 266 entries in OMIM for “schizophrenia” as search term. Further, many of these are large chromosomal regions. It is not only difficult to predict rare pathogenic SVs a priori, but also how to operationalize “around.”
  5. f) For Fig1, the figure legend of 1B does not explain how the ROH region was shown in the plot. The sanger traces of each individual should be displayed.
    We apologize for the vagueness. We have now detailed coding of ROH in text and Figure 1B legend. Sanger sequences have been added to the supplementary information (Supplementary Table 4).
  6. g) In the supplementary SNV discovery section, what is the IGV manual inspection criteria for good, possible, and bad?
    We have deleted these criteria, and simply report SNVs that pass criteria overall.
  7. h) 4.4, paragraph 4, ‘neurodevelopmental disease’ might be more specific than ‘neuroscience.
    We concur and so edited.
  8. i) Typical parameters in WES are shown in ST1’, the ST1 does not seem to have the parameters.
    This was a typographical error and was corrected.
  9. j) Section 2.3.1, paragraph 3, ‘... based on examination of 800 chromosomes.’ Does the author mean 800 individual exome sequencing data/ genome? If this is an internal database, the experiment to construct the database needs to be clarified.
    We clarify now that DNA of 400 subjects was examined (800 chromosomes). These are unrelated, unaffected participants from Pakistan. Note, this was done to confirm that we are not looking at a local polymorphism. We make no attempt at estimating allele frequency from such data.

h)Section 2.3.2 paragraph 1, ‘No SNVs categorized as “good” fit the genetic models.’ the author needs to define what is a good fit and the genetic model tested.
We have removed the term “good and added reference to the models examined in the Methods (2.2)”

__________________________________________________________________________________

Reviewer #2

  1. a) A major drawback of studying SNV in psychiatric patients, especially those detected using family samples or population isolates, is that even though they still may provide useful information about disease pathology, these are private to the family they were identified and thus potentially of limited importance to patients from the general population. It would have been useful if the authors had attempted to place their findings in the context of the currently existing WES sequencing studies, such as those from the SCHEMA consortium.

All methods have both advantages and disadvantages. The GWAS excels at finding risk variants in the general population, but suffers from a 5% false positive rate and difficulty identifying causal genes for elucidating pathophysiology and clinical translation. The approach used here excels at finding potentially causal VOIs that need further study to establish causality (for pathophysiology and clinical exploitation). However, the variant (because of probable genetic homogeneity) applies only to the specific family examined. We do not expect to find another homozygous PSY01 variant [MAF < (10)-5]. We expect to estimate the false positive rate as more families are studied. The PSY01 variant is not in the SCHEMA database and has never been associated with a psychiatric disorder.

  1. b) Following on my first comment, have the authors attempted to investigate existing RNA-seq and proteomic studies to investigate if the specific variant appears to be an eQTL (pQTL) for the USP53 gene? While I understand that such interrogation may not reveal conclusive evidence for the importance of this gene in the etiology of SZ, it still nonetheless would have been informative to confirm (or not) the involvement of this gene in SZ.
    We have not done RNA-seq or proteomic studies. As noted above (and a limitation of the study), the approach does not lend itself well to population-level analyses, and the PSY01 variant is so rare. There is no prior evidence this gene is involved in schizophrenia. However, we note that GRIA2 and GRIP2 interact with USP53; there is considerable prior evidence for involvement of GRIA2 and GRIP2 in schizophrenia.

  1. c) The authors show that USP53 is expressed in mouse hippocampus and co-immunoprecipitates with GRIP2 and GRIA2 proteins, therefore, the authors suggest that USP53 may be a part of the AMPA synapse. While this is valid conjecture, additional studies are needed to verify this.

We agree completely. Although not part of this study, current examination of dissociated rat hippocampal neurons in tissue culture indicates that about 60% of spines with PSD95 colocalize with USP53. Clearly, many more studies are needed and hope for funding them.

“Overall, the study is interesting as it implicates yet another unexplored genetic mechanism in the etiology of SZ.     “

__________________________________________________________________________________

Reviewer #3

1) “The Methods section appears after the Results, which hampers comprehension. A brief outline of the study design and pertinent analysis details, integrated within the text where appropriate, would improve clarity.”
The section on Methods has been moved as suggested.

2)The section 2.3.1 is confusing. The rationale and details for identifying the three chromosomal regions are not clear to me…”
(section 3.3.1 in revision) The new diagram clarifies the rationale (Supplementary Figure)3). In brief, the variants were analyzed by homozygosity mapping and concordance for the affected and discordance for the non-affected. These were filtered to exclude common variants, indigenous polymorphisms, synonymous or in silico undamaged, presence in normative databases, evolutionarily unconserved, etc. The analysis began with homozygosity mapping and filtering to identify variants of interest. No a priori assumptions were made to avoid bias (and no existing data could have pointed to USP53). Seven homozygous regions surfaced (Supplementary Table 3) were present in the affected but not in unaffected. Of these, the large run of homozygosity containing the only one variant, USP53 p.C228R, was predicted as likely pathogenic and occurring in a residue that is highly conserved across vertebrates.

3) Could you clarify why intronic regions are considered a limitation, given that the paper focuses on high-impact variants and specifically emphasizes exonic variants in Section 2.3.1?” We now clarify that as compared to WGS, intronic variants would be missed.

The comment was deleted for clarity.

4) The authors should introduce a variant-filtering step post-variant calling and outline this in the Methods section. For instance, candidate variants for schizophrenia could be those present in affected cases but absent or heterozygous in other family members. The CNV present only in an unaffected family member seems incongruent with the study's hypothesis. Additionally, the term "three samples" is ambiguous and requires clarification.
The Methods now contain the requested information. The reference to three samples is clarified by parenthesis. The CNV in the unaffected family member was included for thoroughness (i.e., what CNVs were detected using the pipeline?), and why were they excluded (not consistent with segregation) from consideration.

5) The focus on USP53 expression in the mouse hippocampus appears hypothesis-driven. While the hippocampus and brain in schizophrenia is acknowledged, the paper would benefit from additional evidence supporting the importance of USP53 in the human brain or hippocampus, potentially sourced from existing literature or public databases.

We emphasize there is no such existing literature or availability in public databases. This is the first demonstration of USP53 expression in the murine brain. As an aside, new unpublished data indicates that 60% of isolated hippocampal neurons show colocalization of USP53 with PSD95 (a post-synaptic marker) in spines. Therefore, USP53 is a new synaptic protein arising from the genetic data.

6) As prioritizing VOIs may introduce bias into the findings, it would be prudent to include a comprehensive list of VOIs in the supplementary materials. The candidate VOIs found through homozygosity mapping are in Supplementary Tables 4-6. The reported USP53 variant was the only VOI that met the stated a priori criteria to prevent bias.

7) “Since there was no prior information about USP53 related to neuroscience or psychiatric disease.” Please note that USP53 is one of ubiquitin-specific peptidases. And ubiquitin-specific peptidases have been implicated in neurodegenerative disorders.
We completely agree, and this is a key area for future research. The citation about a recent review on this topic is # 49 in the bibliography. Of note, USP53 lacks deubiquitinase activity and must act through protein-protein interactions.

8) Please cite the sources for the 3% prevalence rate of psychosis and specify the population from which this prevalence was derived.

Now in first sentence of Introduction.

9) To enhance readability for a general audience, consider defining Loss-of-Function (LOF) variants upon their first mention.
Now included when abbreviation is first mentioned (3rd paragraph of Introduction).

10).The sentence "Both common and rare risk genes associated with schizophrenia were reviewed recently [8]" seems superfluous unless its relevance is clarified.
It is clarified that the reference summarizes the antecedent paragraph and provides many more examples.

11) Unnecessary abbreviations should be avoided for clarity. For instance, "ST" for supplementary table is not intuitive.

Abbreviations such as “ST” and “SF” are now written out for clarity.

_________________________________________________________________________________________________________________

Reviewer 2 Report

Kanwal and colleagues report the identification of a novel, potentially pathogenic, SNV in the USP53 gene in a consanguineous family from Pakistan. The study is well-written and of potential interest to medical professionals. My comments are listed below.

1.       A major drawback of studying SNV in psychiatric patients, especially those detected using family samples or population isolates, is that even though they still may provide useful information about disease pathology, these are private to the family they were identified and thus potentially of limited importance to patients from the general population. It would have been useful if the authors had attempted to place their findings in the context of the currently existing WES sequencing studies, such as those from the SCHEMA consortium.

2.       Following on my first comment, have the authors attempted to investigate existing RNA-seq and proteomic studies to investigate if the specific variant appears to be an eQTL (pQTL) for the USP53 gene? While I understand that such interrogation may not reveal conclusive evidence for the importance of this gene in the etiology of SZ, it still nonetheless would have been informative to confirm (or not) the involvement of this gene in SZ.

3.       The authors show that USP53 is expressed in mouse hippocampus and co-immunoprecipitates with GRIP2 and GRIA2 proteins, therefore, the authors suggest that USP53 may be a part of the AMPA synapse. While this is valid conjecture, additional studies are needed to verify this.

Overall, the study is interesting as it implicates yet another unexplored genetic mechanism in the etiology of SZ.     

Author Response

(The authors gave the same response as above.)

Reviewer 3 Report

The present study employed a consanguineous family, featuring members diagnosed with schizophrenia, to explore rare variants associated with this condition. While the identification of such rare risk variants is undoubtedly important for elucidating the pathology of schizophrenia, I encountered some challenges in comprehending the study design and the details of the methodology. Please find my specific comments below.

1. The Methods section appears after the Results, which hampers comprehension. A brief outline of the study design and pertinent analysis details, integrated within the text where appropriate, would improve clarity.

2. The section 2.3.1 is confusing. The rationale and details for identifying the three chromosomal regions are not clear to me, even after reading the Methods section. The draft seems to indicate that the USP53 variant was identified first, followed by the discovery of three regions related to schizophrenia. Among these 3 regions, only one USP53 variant was found affecting an amino acid. The Method section currently lacks clarity on why a region-based analysis was employed and how the specific regions were determined. Meanwhile, a flowchart summarizing the study design could be an invaluable addition.

3. “WES can suffer from limitations: exon capture, non-coding intronic regions, and validation to detect structural variants.” Could you clarify why intronic regions are considered a limitation, given that the paper focuses on high-impact variants and specifically emphasizes exonic variants in Section 2.3.1?

4. “Two CNVs were present in all three samples and mapped to chromosome 4 (e.g., SF2). One 2.5 kb deletion was detected on chromosome 5, but it occurred only in the father. A 427 kb deletion located on chromosome 9 was found only in one of the affected daughters.” The authors should introduce a variant-filtering step post-variant calling and outline this in the Methods section. For instance, candidate variants for schizophrenia could be those present in affected cases but absent or heterozygous in other family members. The CNV present only in an unaffected family member seems incongruent with the study's hypothesis. Additionally, the term "three samples" is ambiguous and requires clarification.

5. The focus on USP53 expression in the mouse hippocampus appears hypothesis-driven. While the hippocampus and brain in schizophrenia is acknowledged, the paper would benefit from additional evidence supporting the importance of USP53 in the human brain or hippocampus, potentially sourced from existing literature or public databases.

6. As prioritizing VOIs may introduce bias into the findings, it would be prudent to include a comprehensive list of VOIs in the supplementary materials.

7. “Since there was no prior information about USP53 related to neuroscience or psychiatric disease.” Please note that USP53 is one of ubiquitin-specific peptidases. And ubiquitin-specific peptidases have been implicated in neurodegenerative disorders.

Minor comments:

1. Please cite the sources for the 3% prevalence rate of psychosis and specify the population from which this prevalence was derived.

2.To enhance readability for a general audience, consider defining Loss-of-Function (LOF) variants upon their first mention.

3.The sentence "Both common and rare risk genes associated with schizophrenia were reviewed recently [8]" seems superfluous unless its relevance is clarified.

4.Unnecessary abbreviations should be avoided for clarity. For instance, "ST" for supplementary table is not intuitive.

Please improve the clarity, especially for the study design, as I mentioned above.

Author Response

(The authors gave the same response as above.)

Round 2

Reviewer 2 Report

I have no more suggestions for the authors.